# Lockdown Effect on Elderly Nutritional Health

**DOI:** 10.3390/jcm10215052

**Published:** 2021-10-28

**Authors:** Jeyniver Ghanem, Bruno Colicchio, Emmanuel Andrès, Bernard Geny, Alain Dieterlen

**Affiliations:** 1Faculté de Médecine, Université de Strasbourg, 4 Rue Kirschleger, FR-67085 Strasbourg, France; emmanuel.andres@chru-strasbourg.fr (E.A.); bernard.geny@chru-strasbourg.fr (B.G.); 2Institut IRIMAS (7499), IUT, Université de Haute-Alsace, 12 Rue des Frères Lumière, FR-68093 Mulhouse, France; bruno.colicchio@uha.fr (B.C.); alain.dieterlen@uha.fr (A.D.); 3Service de Médecine Interne, Diabète et Maladies Métaboliques, Hôpitaux Universitaires de Strasbourg, FR-67000 Strasbourg, France; 4Service de Physiologie et d’Explorations Fonctionnelles, Hôpitaux Universitaires de Strasbourg, FR-67000 Strasbourg, France

**Keywords:** lockdown, COVID-19, SARS-CoV-2, elderly, bodyweight, malnutrition

## Abstract

Pandemics and lockdowns may be associated with unpremeditated consequences, such as bodyweight changes, isolation, as well as sedentarity. Reports have been published on malnutrition among patients suffering from COVID-19. This study aimed to highlight the short-term effects of the lockdown on the nutritional health of elderly people living at home and benefiting from home care services, yet without any COVID-19 pathology. In 50 subjects displaying weight, body mass index, and MNA score stability two months earlier, we observed significant alterations in these parameters following the lockdown. Thus, malnutrition rose from 28–34% to 58%. Furthermore, trigger factors for malnutrition changed, with social isolation accounting for 64% of the confinement’s deleterious effects among the elderly. In conclusion, despite the elderly being not directly affected by SARS-CoV2, the nutritional status of elderly subjects living at home with no or only mild autonomy loss was greatly and rapidly affected by the lockdown. Moreover, the main trigger factors for malnutrition were essentially related to social isolation and depressive syndromes. Knowing the impact of confinement on the elderly’s health, these results may help further modulate ongoing public health interventions in case of future lockdowns.

## 1. Introduction

Though the pandemic linked to the SARS-CoV2 coronavirus has spread around the world, scientific research is still heavily mobilized to enhance the production of knowledge on the virus to optimize treatment and prevention of this disease. 

Clearly, although without suffering directly from SARS-CoV2, many people are affected by serious deleterious effects in relation with the pandemic. Above all, malnutrition among the elderly is a vicious cycle that alters the person’s state of health, associating weight loss and fragility. This, in turn, translates into increasingly long and common pathological episodes [1,2]. 

The risk of malnutrition is multifactorial in nature, including psychosocial aspects, autonomy loss, decompensation of chronic pathologies, dementia syndromes, oral disorders, poly-medications, and other medical conditions. There are multiple consequences, including increased mortality and morbidity, as well as autonomy loss and deteriorated quality of life. In addition, malnutrition is associated with its own morbidity and mortality in the elderly, which is independent of the underlying disease, and exerts a negative impact on their quality of life. It is, therefore, essential to prevent malnutrition or, in a worst-case scenario, to manage this condition as soon as diagnosed.

Although COVID-19 is expected to increase malnutrition, as well as stunting, and micronutrient deficiencies [3], only relatively few data [4] have been reported concerning the nutritional status in elderly subjects during the COVID-19 pandemic time.

This study, therefore, thought first to assess the impact of COVID-19-induced lockdown on the nutritional status of elderly subjects living at home, with none or only mild loss of autonomy. The study’s second objective was to examine the origins of such impairment.

## 2. Population and Methods 

In order to detect and prevent undernutrition in the elderly living at home, the service e-Nutriv (available online: https://www.apamad.fr/e-nutriv-detecter-risques-de-denutritions-personnes-agees/ accessed on 19 August 2021), supported by the APA (Réseau APA is a non-profit network of the Social and Solidarity Economy, which intervenes in the social, medico-social, and health fields to maintain home help and care in Haut-Rhin) network, recruited a cohort of 300 people, who were living in France and receiving help at home, involving monthly monitoring of weight and MNA^®^ scoring. The criteria of inclusion were an age of at least 65 years, living autonomously (without the help of the family) at home, as well as benefiting from home care services with the APA network. Due to the pandemic, a complete stop (for the entire 300 subjects) of the one-month follow-up was implemented due to the lockdown (March 2020). For fifty subjects, follow-up was resumed one month later (April 2020), and this sub-cohort served as the basis for our analysis. This sample of fifty persons has remained stable in regards to the two measurements performed before the March 2020 lockdown in France. These fifty subjects (gender = 28% M, 72% F; age = 84.6 ± 6.7 years) signed a written informed consent jointly with their families. This consent form was approved by the Ethical Committee of the Strasbourg University (3 December 2020, CE-2020-197).

As proposed by the *Haute Autorité de Santé* (HAS) in 2007 [5] and the European Society for Clinical Nutrition and Metabolism (ESPEN) in 2017 [6], the subjects’ nutritional health was determined by following weight, body mass index (BMI), and MNA^®^ score (MNA^®^ Short Form, 2006 Society of Products Nestlé S.A. MNA is a validated nutrition screening and assessment tool that can identify geriatric patients aged 65 and above who are malnourished or at risk of malnutrition) [7] variations from prior to the lockdown (two times one-month apart, so as to insure the parameters’ stability) to after the lockdown [8]. Indeed, most people who are malnourished lose weight—however, such people may be at a healthy weight or even overweight, yet still be malnourished. To illustrate, this may occur when people have insufficient nutrients at their disposal through their diet, including certain types of vitamins and minerals. On the other hand, if people lose 5% of their body weight within 1 month, they may be malnourished. Once malnutrition had been established, specific trigger factors were searched for, per anamnesis assessment. In the absence of some understandable trigger factors, further investigations were required among the patient’s family circle.

The e-Nutriv project deploys connected objects, such as connected tablets or weighing pads, which allowed for notifying alerts to medical, family, and professional circles, in addition to monitoring their situation. All the data collected were anonymized and analyzed using the statistical software R (https://www.r-project.org/, accessed date 19 August 2021). The paired binomial test and symmetry tests were performed for malnutrition alert analysis. A paired *t*-test was applied to assess malnutrition indicator variations. A Spearman’s test was carried out to assess the potential relationships between weight and MNA^®^ Score. Statistical significance was set at 0.05.

## 3. Results and Discussion 

Considering the entire population, malnutrition was evaluated twice before the lockdown, with a prevalence of 28% and 34% recorded in January and February 2020, respectively. This is entirely in line with previous data demonstrating malnutrition prevalence between 25–29% for the elderly living at home and benefiting from home care services [9].

Once the confinement was established at the beginning of March (11 March 2020), most home care services were suspended, and malnutrition suddenly rose to 58%.

Paired binomial testing for alert tests and symmetry tests of alert tables revealed a significant switch from normal nutrition to a malnourished status, with values of *p* = 0.03 and *p* = 0.02 for February 2020 and April 2020, respectively. 

Importantly, weight, BMI, and MNA^®^ scores had been stable before the lockdown, whereas they significantly decreased during the lockdown (*p* = 0.0002 for weight, *p* = 0.0001 for BMI, and *p* = 0.0000 (<0.001) for MNA^®^ scores) (Table 1 and Figure 1, Figure 2 and Figure 3).

To better understand our subjects’ characterization, we considered a weight loss of ≥5% to be of concern, which concerned 11 persons versus 39 (22%) (Figure 4).

Nevertheless, no significant difference was observed between these two groups in terms of gender, age, MNA^®^ score, and mobility.

However, BMI was strongly and positively correlated with weight (r = 0.0003, *p* = 0.000), whereas the MNA^®^ score was positively, yet not strongly, correlated with weight (r = 0.22; *p* = 0.008). 

Our second aim was to investigate the mechanisms potentially involved. To this end, we analyzed known risk factors leading to malnutrition changes, before and after the lockdown. The percentage of the malnutrition’s trigger factors among our population was added to the table established by the HAS in 2007 [5]. We have emphasized in red the main cause of each factor, as detected in our study population (Figure 5).

Interestingly, before the confinement, the malnutrition triggers were mainly dementia and dependency, each of them contributing to 30% of the total malnutrition causes. This was followed by psycho-socio-environmental factors, involving mainly social isolation, as well as acute disorders such as infectious disease, with 17 and 13% contributions, respectively (Figure 5).

After the confinement, about 64% of the malnutrition cases appeared to be related to psycho-socio-environmental factors, the second cause being psychiatric troubles, resulting in a 13% contribution. For the third place, the factors contributing to malnutrition were mobility dependence and acute infections, with a 7% contribution for each (Figure 6).

These results clearly demonstrated that, despite acute infections being markedly reduced on account of the confinement, social and psychological factors were clearly enhanced among the studied population.

This is particularly significant. Indeed, when analyzing social connections with variables that predict a person’s chances of survival or death, social relationships, including family, friends, and neighbors, were shown to increase our chance of survival by 50% [10]. Thus, “low social interaction” may be equivalent to smoking 15 cigarettes a day, whereas alcoholism seems to be more harmful than sedentarily, being twice as high as the risk associated with obesity [10]. Overall, the influence of both objective and subjective social isolation on risk for mortality is comparable with well-established mortality risk factors [11].

In brief, we summarize the prevention of malnutrition by four specific terms, involving promotion, enhancement, information, and training. Of course, it is absolutely essential to promote and enhance prevention measures, usually by means of communication with regard to the means available, achievement and pursuit of specific objectives, and contribution of each item to risk protection.

In contrast, the treatment of malnutrition often involves more individualized approaches. The standard treatment of undernutrition aims to achieve an optimal protein and energy supply according to the patient’s specific needs, so as to reduce the effects of catabolism, and minimize the loss of the body’s protein mass.

## 4. Conclusions

Even if not struck directly by SARS-CoV2, the nutritional status of elderly subjects living at home with no or only mild autonomy loss has been greatly and rapidly affected by the lockdown established in France and almost all countries around the world. Importantly, the main trigger factors for malnutrition in the elderly were thereby altered, with social isolation accounting for 64% of the confinement’s deleterious effects.

We are aware that the study’s population size was too small for good-quality statistics, given that out of the 300 people initially recruited into the cohort, only 50 completed the study with good quality data. From a quantitative analysis, the study has thus been condensed to a qualitative analysis on account of these circumstances.

These original data should be taken into serious consideration before any political decision is made concerning a new future confinement—moreover, actions should already be planned and implemented to further reduce social isolation of the elderly that are in enough good health to be able to stay at home.

Further perspectives could involve determining whether such e-Nutriv follow-up, which considers weight, BMI, and MNA^®^ scoring, would be instrumental in early detecting and diminishing malnutrition among elderly people living at home.

## Figures and Tables

**Figure 1 jcm-10-05052-f001:**
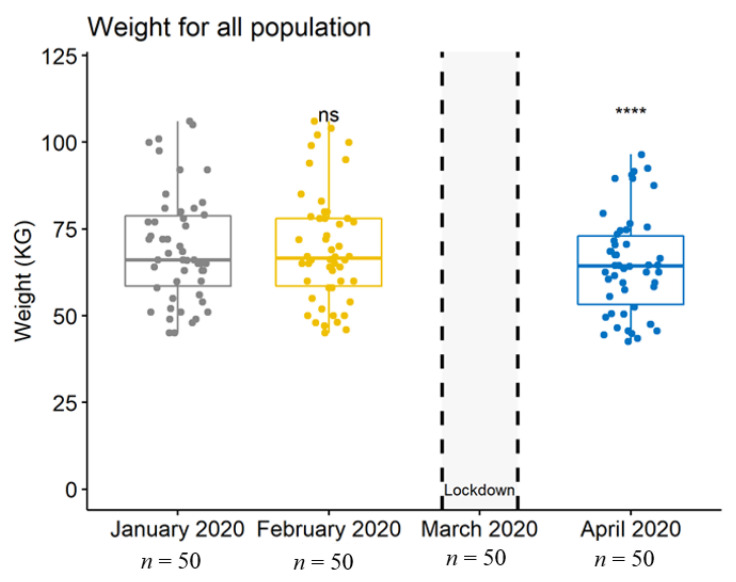
Boxplot depicting the population’s weight at each data collection. ns, no significant difference between January 2020 and April 2020; ****, highly significant difference between January 2020 and April 2020.

**Figure 2 jcm-10-05052-f002:**
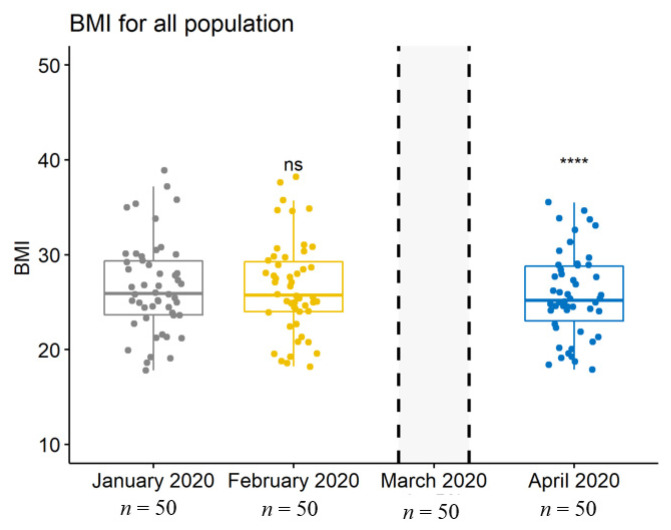
Boxplot depicting the population’s body mass index (BMI) at each data collection. ns, no significant difference between January 2020 and April 2020; ****, highly significant difference between January 2020 and April 2020.

**Figure 3 jcm-10-05052-f003:**
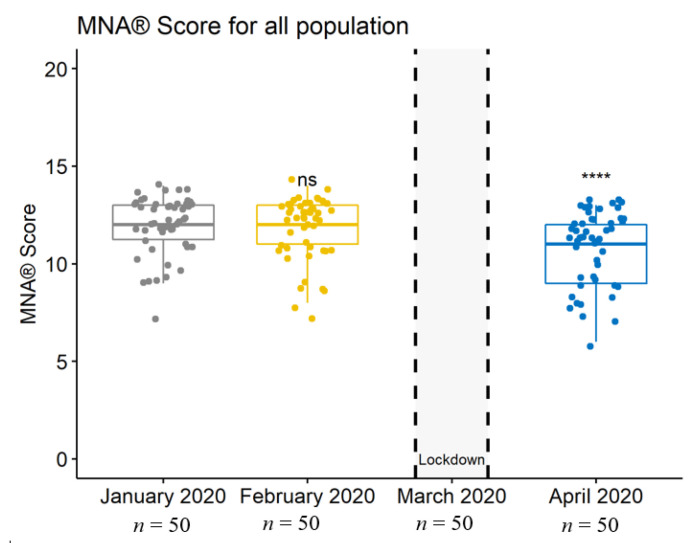
Boxplot depicting the population’s MNA^®^ score at each data collection. ns, no significant difference between January 2020 and April 2020; ****, highly significant difference between January 2020 and April 2020.

**Figure 4 jcm-10-05052-f004:**
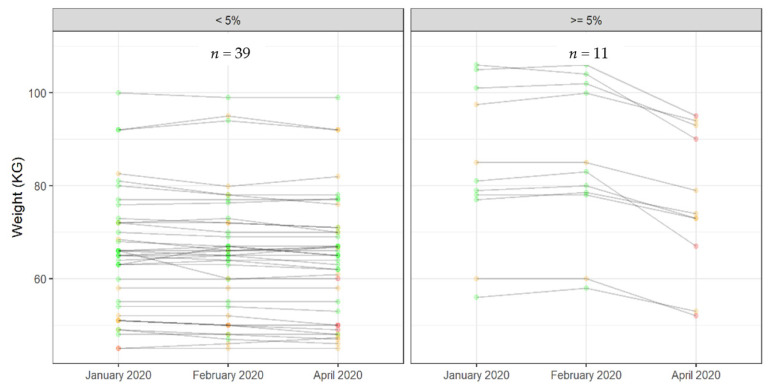
Individual weight variation <5% and ≥5% after lockdown.

**Figure 5 jcm-10-05052-f005:**
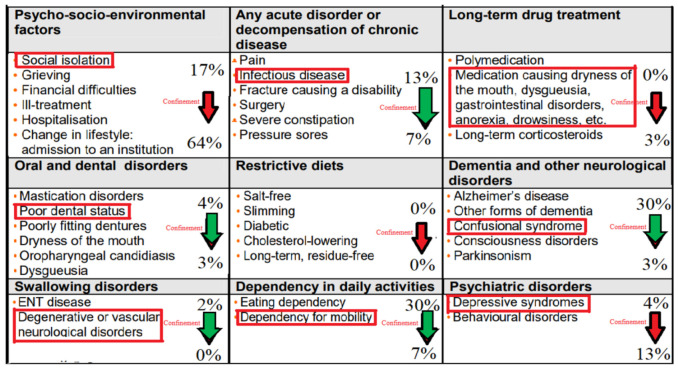
Trigger factors for malnutrition in the elderly, and percentage of prevalence before and after the confinement.

**Figure 6 jcm-10-05052-f006:**
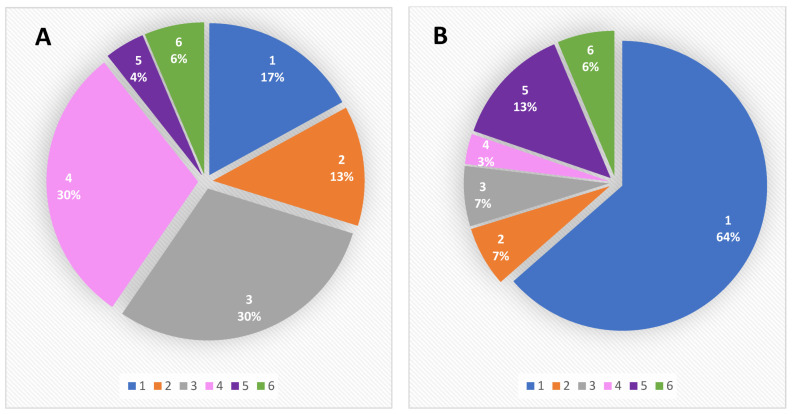
Malnutrition causes before (**A**) and after the lockdown (**B**). 1. Social Isolation; 2. Infection Disease; 3. Dependency for Mobility; 4. Confusional Syndrome; 5. Depressive Syndromes; 6. Others (Oral and Dental Disorders; Swallowing Disorders; Restrictive Diets; Long Term Drug Treatments).

**Table 1 jcm-10-05052-t001:** Mean of the three alert criteria depending on time.

	Weight (kg)	BMI	MNA^®^ Score
January 2020 (*n* = 50)	69.63 ± 15.97 ^a^	26.61 ± 4.83 ^a^	11.98 ± 1.53 ^a^
February 2020 (*n* = 50)	69.51 ± 16.24 ^a^	26.56 ± 4.91 ^a^	11.86 ± 1.58 ^a^
April 2020 (*n* = 50)	67.28 ± 14.49 ^b^	25.75 ± 4.49 ^b^	10.82 ± 1.91 ^b^

^a^ Group 1; ^b^ Group 2. BMI: body mass index.

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
