# Peer review of "Lockdown Effect on Elderly Nutritional Health"

_jcm, 2021, doi:10.3390/jcm10215052_

Round 1
Reviewer 1 Report
The authors should precise in their title which impact they aim to assess in elderly patients (malnutrition?).
The introduction: the authors should better describe the impact of malnutrition in elderly patients.
Results:
A flow-chart would be of help to understand the recruitment of the patients and those who did not benefit from home care services during the lockdown.
It is still unclear to me if home care services were suspended for all patients. I would also like to know more about the studied population (at baseline).
For me, the methodology used in table 2 is unclear and should be further discussed in Methods.
I don't see Figures 5 and 6?
The authors should discuss limitations of their study.
Minor comments:
-is it possible to know if patients had other help at home than the home care services (help from the family?)
-please standardize COVID-19 or covid-19 and improve English.
Author Response
Comments from Reviewer #1
- Comment 1: The authors should precise in their title which impact they aim to assess in elderly patients (malnutrition?).
Response: Thank you for pointing this out. We entirely agree with this comment and have thus included the term “Nutritional” in the title.
- Comment 2: The introduction: the authors should better describe the impact of malnutrition in elderly patients.
Response: We would like to thank Reviewer #1 for this remark to which we agree. Accordingly, we have added a paragraph to Introduction section in order to comply with the Reviewer #1’s remark, as follows:
The risk of malnutrition is multifactorial in nature, including psychosocial aspects, autonomy loss, decompensation of chronic pathologies, dementia syndromes, oral disorders, poly-medications, as well as other medical conditions. There are multiple consequences, including increased mortality and morbidity, as well as autonomy loss and deteriorated quality of life. In addition, malnutrition is associated with its own morbidity and mortality in the elderly, which is independent of the underlying disease and exerts a negative impact on their quality of life. It is, therefore, essential to prevent malnutrition or, in a worst-case scenario, to manage this condition as soon as diagnosed.
- Comment 3: A flow-chart would be of help to understand the recruitment of the patients and those who did not benefit from home care services during the lockdown.
Response: We thank Reviewer #1 for this suggestion. Yet, instead of including a study flowchart to help understand patient recruitment, we have rather added some explanatory sentences to the Population and Methods section, as follows:
In order to detect and prevent undernutrition in the elderly living at home, the e-Nutriv[1] service, supported by the APA[2] network, recruited a cohort of 300 people, who were all living in France and receiving help at home, involving monthly monitoring of weight and MNA® scoring. The criteria of inclusion were an age of at least 65 years, living autonomously (without the help of the family) at home, as well as benefiting from home care services with the APA network. Due to the pandemic, a complete stop (for the entire 300 subjects) of one-month follow-up was implemented due to the lockdown (March 2020). For fifty subjects, follow-up was resumed one month later (April 2020), and this sub-cohort of 50 elderly served as the basis for our analysis.
- Comment 4: It is still unclear to me if home care services were suspended for all patients. I would also like to know more about the studied population (at baseline).
Response: Indeed, we can understand Reviewer #1’s confusion concerning the suspension of the home care services. As a result, we have now provided more information on the study population, as already outlined before and restated below:
In order to detect and prevent undernutrition in the elderly living at home, the e-Nutriv[3] service, supported by the APA[4] network, recruited a cohort of 300 people, who were all living in France and receiving help at home, involving monthly monitoring of weight and MNA® scoring. The criteria of inclusion were an age of at least 65 years, living autonomously (without the help of the family) at home, as well as benefiting from home care services with the APA network. Due to the pandemic, a complete stop (for the entire 300 subjects) of one-month follow-up was implemented due to the lockdown (March 2020). For fifty subjects, follow-up was resumed one month later (April 2020), and this sub-cohort of 50 elderly people served as the basis for our analysis.
- Comment 5: For me, the methodology used in table 2 is unclear and should be further discussed in Methods.
Response: Again, we agree with Reviewer #1’s query and we have thus incorporated a few additional sentences to the Population and Methods section, line 69 to 71.
- Comment 6: The authors should discuss limitations of their study.
Response: Reviewer #1’s has raised an essential issue here. Accordingly, we have added a few sentences to the Conclusion section in order to highlight the study’s limitations, as well as the manner we managed these issues, as follows:
We are aware that the study’s population size was too small for good-quality statistics, given that out of the 300 people initially recruited into the cohort; only 50 completed the study with good quality data. From a quantitative analysis, the study has thus been condensed to a qualitative analysis on account of these circumstances.
- Comment 7: is it possible to know if patients had other help at home than the home care services (help from the family?)
Response: We have added the response to this query by providing more details to the Population and Methods section, pointing out that the elderly were autonomous, without any family help, as follows:
In order to detect and prevent undernutrition in the elderly living at home, the e-Nutriv[5] service, supported by the APA[6] network, recruited a cohort of 300 people, who were all living in France and receiving help at home, involving monthly monitoring of weight and MNA® scoring. The criteria of inclusion were an age of at least 65 years, living autonomously (without the help of the family) at home, as well as benefiting from home care services with the APA network. Due to the pandemic, a complete stop (for the entire 300 subjects) of one-month follow-up was implemented due to the lockdown (March 2020). For fifty subjects, follow-up was resumed one month later (April 2020), and this sub-cohort of 50 elderly served as the basis for our analysis.

Reviewer 2 Report
- The introduction should expand on the reasons for giving malnutrition among this population
- From the methods, it is not clear how the causes for giving malnutrition among the population were monitored
- Figure 1 does not show the weight loss of the subjects as seen in Table 1, the graph even looks higher after lockdown
- The discussion lacks support from studies that have linked malnutrition to its causes such as depression, loneliness, etc. in this population
- It is advisable to add to the recommendations for the prevention of malnutrition among this population according to the reasons that cause malnutrition
Author Response
Comments from Reviewer #2
- Comment 1: The introduction should expand on the reasons for giving malnutrition among this population.
Response: First of all, we would like to thank Reviewer #2 for the time and energy spent on reviewing our paper. In line with his/her first query, which was also addressed by Reviewer #1, we have included additional information on malnutrition status of our study population, as follows:
The risk of malnutrition is multifactorial in nature, including psychosocial aspects, autonomy loss, decompensation of chronic pathologies, dementia syndromes, oral disorders, poly-medications, as well as other medical conditions. There are multiple consequences, including increased mortality and morbidity, as well as autonomy loss and deteriorated quality of life. In addition, malnutrition is associated with its own morbidity and mortality in the elderly, which is independent of the underlying disease and exerts a negative impact on their quality of life. It is, therefore, essential to prevent malnutrition or, in a worst-case scenario, to manage this condition as soon as diagnosed.
- Comment 2: From the methods, it is not clear how the causes for giving malnutrition among the population were monitored.
Response: We agree with the issue raised by Reviewer #2 and have thus incorporate a further clarification on this into the Population and Methods section, as follows:
As proposed by the Haute Autorité de Santé (HAS) in 2007 [5] and European Society For Clinical Nutrition and Metabolism (ESPEN) in 2017 [6], the subjects’ nutritional health was determined by following weight, body mass index (BMI), and MNA® score[1] [7] variations from prior to (two times one-month apart so as to insure the parameters’ stability) to after the lockdown [8]. Indeed, most people who are malnourished lose weight; however, such people may be at a healthy weight or even overweight, yet still be malnourished. To illustrate, this may occur when people have insufficient nutrients at their disposal through their diet, such as certain types of vitamins and minerals. On the other hand, if people lose 5% of body weight within 1 month, they may be malnourished. Once malnutrition had been established, specific trigger factors were searched for per anamnesis assessment. In the absence of some understandable trigger factors, further investigations were required among the patient’s family circle.
Comment 3: Figure 1 does not show the weight loss of the subjects as seen in Table 1, the graph even looks higher after lockdown
Response: We thank Reviewer #2 for this remark. Indeed, the weight loss is not easily detected. Given this context, we have now added a blue line to Figure 1, which clearly shows that in April 2020, the average body weight was distinctly lower than in January and February 2020.
- Comment 4: The discussion lacks support from studies that have linked malnutrition to its causes such as depression, loneliness, etc. in this population
Response: We thank Reviewer #2 for this suggestion. In line with this query, we have expanded on these issues and added information relating to Figures 5 and 6, as follows:
Figure 5 and 6: Malnutrition causes before (left) and after the lockdown (right). 1. Social Isolation; 2. Infection Disease; 3. Dependency for Mobility; 4. Confusional Syndrome; 5. Depressive Syndromes; 6.Others: Oral and Dental Disorders; Swallowing Disorders; Restrictive Diets; Long Term Drug Treatments.
- Comment 5: It is advisable to add to the recommendations for the prevention of malnutrition among this population according to the reasons that cause malnutrition.
Response: We are grateful to Reviewer #2 for pointing this out. Therefore, we have added more information to the new manuscript’s version, summarizing the basics of prevention of malnutrition and treatments.
Overall, the influence of both objective and subjective social isolation on risk for mortality is comparable with well-established mortality risk factors [11].
In brief, we summarize the prevention of malnutrition by four specific terms, involving promotion, enhancement, information, and training. Of course, it is absolutely essential to promote and enhance prevention measures, usually by means of communication with regard to the means available, achievement and pursuit of specific objectives, and contribution of each item to risk protection.
In contrast, the treatment of malnutrition often involves more individualized approaches. The standard treatment of undernutrition aims to achieve an optimal protein and energy supply, according to the patient's specific needs, so as to reduce the effects of catabolism and minimize the loss of the body's protein mass.

Round 2
Reviewer 1 Report
The authors have answered must of the reviewers' questions and the manuscript has been improved and is much more easy to read and understand.
The treated topic merits attention in the context of the pandemic.